# A Novel Reticular Retained Austenite on the Weld Fusion Line of Low Carbon Martensitic Stainless Steel 06Cr13Ni4Mo and the Influence on the Mechanical Properties

**Fan Peng [1], Zhourong Feng [2], Yu Zhao [3] and Jianzhou Long [3],***

[1]  Kocel Machinery Co., Ltd., Yinchuan 750021, China; fan.peng@kocel.com
[2]  Kocel Steel Foundry Co., Ltd., Yinchuan 750021, China; zhourong.feng@kocel.com
[3]  Anhui Key Laboratory for High-Performance Non-Ferrous Metal Materials, Anhui Polytechnic University, Wuhu 241000, China; zhaoyu@ahpu.edu.cn
*   Correspondence: jzlong11s@alum.imr.ac.cn; Tel.: +86-0553-2871252

**Abstract:** AbstractWhen performing fluorescent magnetic particle testing in steel welding-repaired zones, as traditional viewpoint usually ascribes magnetic particle indications to discontinuous welding defects such as cracks, incomplete fusion, etc., welding-repaired zones showing indications of these defects are always judged to be unqualified. In this study, a novel reticular phase was identified on the weld fusion line of 06Cr13Ni4Mo. By selected area electron diffraction, it was proved to be an austenite. By roasting test, it was proved to be induced by the thermal effect of welding. Non-ferromagnetic reticular austenite reduces the overall magnetic permeability, leading to the presence of fluorescent magnetic particle indication. Though the mechanical properties of the welding repaired zone are changed by the reticular austenite with the yield strength, tensile strength, and microhardness decreasing to 571 MPa, 752 MPa, and 279, respectively, they still exceed the required values of standard specifications. 06Cr13Ni4Mo welding-repaired zones showing magnetic particle indications induced by reticular austenite are qualified and should be accepted.

**Keywords:** 06Cr13Ni4Mo; welding; austenite; mechanical property

## 1. Introduction

As a typical high-strength and corrosion-resistant stainless steel, 06Cr13Ni4Mo is widely used in hydraulic turbine flow parts [1–3]. These parts are driven by high-speed water flow to generate electricity, so there is generally a high requirement on the mechanical properties of 06Cr13Ni4Mo [3,4]. For example, the industrial standard specifications require that the yield strength $\sigma_s$, tensile strength $\sigma_b$, elongation to failure $\delta_f$, and microhardness have to be no less than 550 MPa, 750 MPa, 15%, and 221~286, respectively [5,6].

In order to meet the mechanical property requirement, special alloy contents and heat treatment are developed for 06Cr13Ni4Mo [7,8]. The Ni content is between 3.5~5 wt.% [8,9]. Ni can enlarge the high-temperature austenite phase zone, increase the phase stability of austenite and the solubility of Cr, and improve the hardenability [8,9]. Normalizing and double-tempering treatment are usually applied with a normalizing temperature and tempering temperature of 950~1050 °C and ~600 °C, respectively [9–12]. After tempering, a mixed microstructure composed of lath martensite and trace reversed austenite dispersing between laths can be obtained [9,13,14]. This dual-phase microstructure endows 06Cr13Ni4Mo with excellent strength and ductility, which exceed the required values in the above standard specifications [5,6].

There are usually various solidification defects, such as shrinkage and porosity, in 06Cr13Ni4Mo castings [15]. For safety, these defects must first be dug out, and then welding repair is used to fill the cavities left by the digging [12,16–19]. After welding repair, to ensure the quality of the welding, nondestructive testing (NDT) is always performed. The

NDT techniques usually applied are penetration testing (PT), ultrasonic testing (UT), and fluorescent magnetic particle testing (FMPT) [20]. Because FMPT is suitable to complex workpiece examination and quickly displays the position and size of the defect, it is becoming the most widely used NDT technique [21]. During a cycle of FMPT, the workpiece is sprayed with magnetic particle suspension and simultaneously energized by a magnetic field. The magnetic field at defects leaks out of the workpiece surface. Thus, fluorescent magnetic particles in the suspension accumulate on the site of leakage. The accumulation induces fluorescent magnetic particle indication (FMPI) under the illumination of 365 nm ultraviolet light [20,22].

As discontinuous welding defects, such as cracks and pores, lead to severe magnetic flux leakage, FMPI is always ascribed to these discontinuous welding defects in practical applications. As a result, welding-repaired zones showing FMPI are always judged to be unqualified [23–26].

It is well known that the welding process is very complicated [12,27]. The non-equilibrium solidification conditions found during fusion welding promote the formation of metastable and various phases depending on the material being processed [28,29]. Moreover, the special alloy content adds complexity to the microstructure after welding. However, as FMPT is scarcely performed in materials research labs of universities, for 06Cr13Ni4Mo welding-repaired zones, there are still no investigations on the influence of the complex welding microstructure on FMPI. In fact, FMPI was found in welding-repaired zones without discontinuous defects for 06Cr13Ni4Mo big castings in Kocel steel foundry. It was speculated that these FMPIs were induced by the complex welding microstructure. That is the origin of this study.

In this paper, microstructures of 06Cr13Ni4Mo weld fusion line were characterized. The origin of FMDI around the welding areas was studied. Formation mechanism of a novel reticular phase and influence on mechanical properties of 06Cr13Ni4Mo welding repaired zones were analyzed and discussed, hoping to clarify the FMDT result evaluation of 06Cr13Ni4Mo castings.

## 2. Experimental Methods

The material used in this study was 06Cr13Ni4Mo (chemical composition in Table 1) in a heat-treated state. The heat treatments used were normalizing and tempering, with temperatures of 1030 °C and 600 °C, respectively. To verify the repeatability of FMPI in welding-repaired zones, the welding-repair process of defects in large 06Cr13Ni4Mo castings was simulated. The simulation was performed using large test blocks (160 mm × 160 mm× 60 mm with a U-shaped groove ($\Phi$100 mm× 30 mm) (Figure 1a). The widely used manual metal inert-gas (MIG) welding was used, and the welding parameters are listed in Table 2. The welding parameters were chosen to be the same as those used for welding repair in Kocel steel foundry. As the thickness of each welding layer is about 2 mm, it takes about 15 layers to fully fill the U-shaped groove. Taking the time for interlayer cooling into account, it takes about 20–30 min for the welding repair of the groove by fusion welding.

**Table 1.** Chemical composition of 06Cr13Ni4Mo.

| Element | C | Si | Mn | P | S | Cr | Ni | Mo | Cu | V |
|---|---|---|---|---|---|---|---|---|---|---|
| Content (wt.%) | 0.07 | 0.88 | 0.63 | 0.011 | 0.008 | 12.4 | 4.1 | 0.58 | 0.28 | 0.13 |

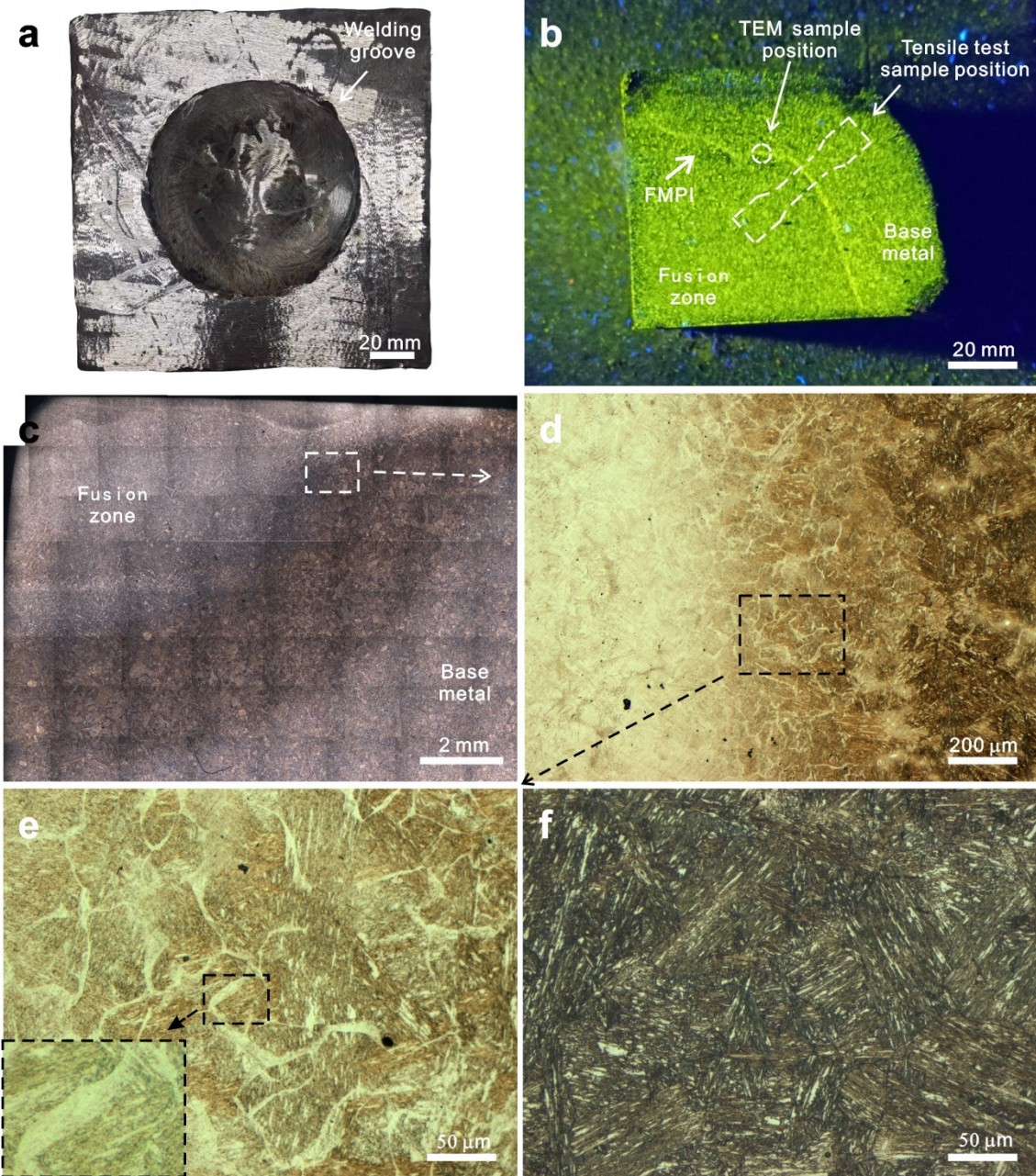

**Figure 1.** (**a**) The workpiece with the U-shaped groove used for welding test. (**b**) Fluorescent magnetic particle indication on the welded workpiece. (**c**) Optical micrographs of the macrosection of the welding repaired zone. (**d**) High magnification optical micrographs of the weld fusion line. (**e**) High magnification optical micrographs of the aera indicated by the black dotted box in (**d**). (**f**) Optical micrographs of base metal. Note that, for convenience of fluorescent magnetic particle testing, only 1/4 part of the welded workpiece was used in (**b**). The position of TEM samples and tensile test samples of the materials after welding are indicated in (**b**).

**Table 2.** Process parameters of MIG welding used in this study.

| Process Parameter | Value | Process Parameter | Value |
|---|---|---|---|
| Preheat Temperature | 120 °C | Interpass Temperature | 180 °C |
| Voltage | 30 V | Welding wire | 410 NiMo |
| Current | 240 A | Shielding gas | 95% Ar + 5% $CO_2$ |
| Travel speeding | 300 mm/min | Position of groove | Flat |

To simulate the microstructure transformation in the welding heat-affected zone (HAZ), 06Cr13Ni4Mo blocks with dimensions of 35 mm × 35 mm × 135 mm were roasted on the top surface. During roasting, the block was placed on the ground, and the top surface was roasted by acetylene flame for 30 min. During roasting, the flame nozzle was kept 150 mm away from the top surface. Using an infrared temperature gun, the temperature in the block was found to be constant after 12 min, and the gradient temperature distribution in the block can be calculated using the following discoveries and assumptions: (1) the top layer with a thickness of ~7 mm melted during the roasting; (2) the stable temperature at the bottom was measured to be 336 °C by the temperature gun; (3) heat dissipation of the block is mainly realized by radiation and follows the Stefan–Boltzmann law in Equation (1).

$$E = \sigma T^4 \tag{1}$$

where $E$, $\sigma$, and $T$ are the radiant force, the Stefan–Boltzmann constant, and the temperature of the block, respectively.

When the stable state is reached after 12 min, the relation between $T$ and $x$ (distance from the roasted top surface) can be given as follows:

$$T = 5025x^{-\frac{2}{3}} + 141 \tag{2}$$

Samples used for microstructure characterization and mechanical testing were prepared by wire electrical discharge machining (Harbin Welding Institue Co., Ltd., Harbin China). The metallographic and X-ray diffraction (XRD) samples were sequentially processed by mechanical grinding, mechanical polishing, and chemical etching. The etching solution was 5g $FeCl_3$ + 50 mL HCl + 100 mL distilled water. Keens VHX-5000 (Keyence corporation, Shanghai China) and Bruck X-ray diffractometer D8 Advance (Bruker AXS, Karlsruhe, Germany) were used for metallographic observation and XRD. FEI Tecani F30 transmission electron microscope (TEM) (FEI Company, Hillsboro, USA) was used for microstructure characterization. The position of the TEM sample was indicated by the white dashed circle in Figure 1b. Scanning electron microscopy (SEM) was performed using JEOL JSM 7100F equipped with an Oxford X-Max energy dispersive spectrometer (EDS) (JEOL CO., Ltd., Tokyo, Japan) at a voltage of 15 kV. The microhardness was measured using Qness Q10A+. The load was 100 g and the holding time was 10 s, a widely used value. Instron 5982 tensile testing machine (The Hong Kong Polytechnic University, Hong Kong, China) was used for tensile testing at room temperature. Dog-bone-shaped tensile samples with a gauge section of 6 × 2 mm$^2$ and a gauge length of 15 mm were prepared. The position of the tensile test samples is indicated by the white dashed box in Figure 1b. Clip-on extensometers (Instron 2620-601) (Instron corporation, Boston, USA) were used for measuring of the engineering strain. The strain rate was $10^{-3}$ s$^{-1}$. For each kind of the materials, three dog-bone-shaped tensile samples were prepared and tested.

## 3. Results

### 3.1. Microstructures of the Fusion Line

Figure 1 shows FMPT results of the welding-repaired block and microstructures around the weld fusion line. Due to the large size of the welding-repaired block, and for convenience of FMPT, only one-quarter was cut down and used for FMPT. There is

apparent FMPI surrounding the fusion zone (Figure 1b), which is consistent with the FMPT result of the welding-repaired zone of 06Cr13Ni4Mo big castings in Kocel steel foundry. As FMPI is usually considered an indication of discontinuous welding defects, such as hot cracks, porosity, and incomplete fusion [3,15,24], and these discontinuous defects always result in severe mechanical property decrease, these big casting products may be rejected by customers. The origin of the FMPI must be revealed.

Using mechanical polishing, materials of FMPI zones were removed layer by layer. No discontinuous defects were observed. However, metallographic observations show that, compared with the tempered martensite of the base material (Figure 1f), an abnormal white phase with a width of about several microns formed on the fusion line (Figure 1c,d). It distributes on the boundaries of the martensitic clusters and forms a network-like structure (Figure 1e). High magnification observation shows that the tempered lath structure disappeared in the white phase (Figure 1e). Compared with the tempered lath martensite of the base metal, the fusion line shows a mixed structure consisting of reticular white phase and tempered martensite. The reticular white phase shows special morphology and has not been reported previously [9,12,27].

Figure 2 shows the SEM image and EDS spectrum of the reticular phase on the weld-fusion line. In contrast to the lath morphology of the surrounding martensite, no apparent laths can be observed in the reticular phase (Figure 2a), which is consistent with that shown by the optical micrographs (Figure 1e). The EDS spectrum shows that the Ni content in reticular phase is 5.1 at.% (Figure 2b), corresponding to the weight percentage of ~5.9 wt.%. Compared with the average concentration of Ni (4.1 wt.%) in the base material (Table 1), it can be found that Ni enrichment occurred in the reticular phase. For 06Cr13Ni4Mo, Ni enrichment is frequently detected in the reversed austenite by low temperature tempering, and the enriched content is usually 7–8 wt.% [30–32]. The Ni content (5.9 wt.%) observed here in the reticular phase is a little lower than that (7–8 wt.%) in the reversed austenite by low-temperature tempering. The potential reasons for this will be analyzed in the discussion.

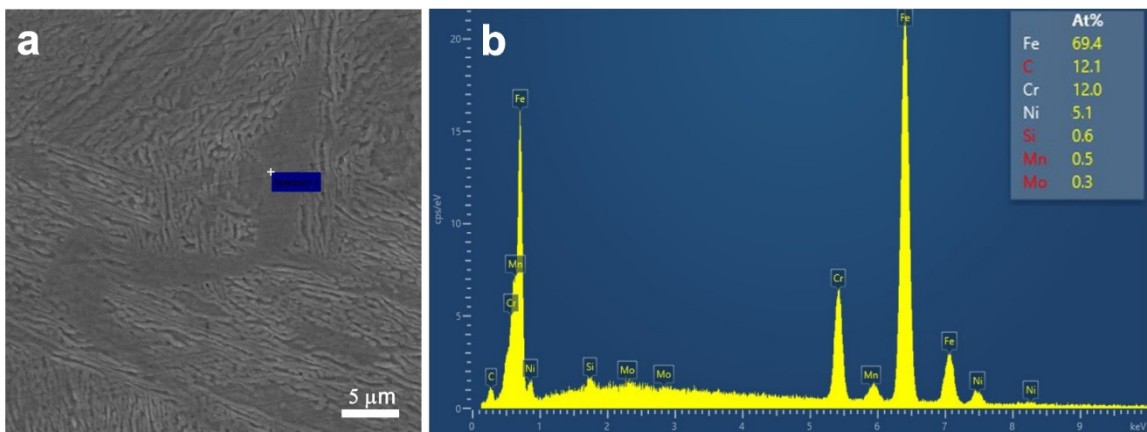

**Figure 2.** (**a**) SEM images of the reticular phase on the weld fusion line. (**b**) EDS spectrum at the location indicated by the cross symbol in (**a**).

Figure 3 shows TEM microstructures of the fusion line. In the scanning transmission mode, most of the area is tempered lath martensite. Near the triple junction of martensitic cluster boundaries, the lath structure disappeared. This area is about 5–10 μm wide (Figure 3a). As opposed to the surrounding lath structure, this area shows no obvious contrast, indicating that substructures have disappeared (Figure 3b). Judging by the distribution, size, and structural characteristics, it can be deduced that phases near this triple junction should be the same reticular white phase observed in Figure 1d,e. Figure 3c shows the selected aera electron diffraction near the triple junction. Compared with the standard diffraction pattern of face-centered cubic structure, the crystallographic plane

indices can be marked properly. The [1 1 0] interplanar spacing was calculated to be 0.341 nm, demonstrating that it is face-centered cubic austenite.

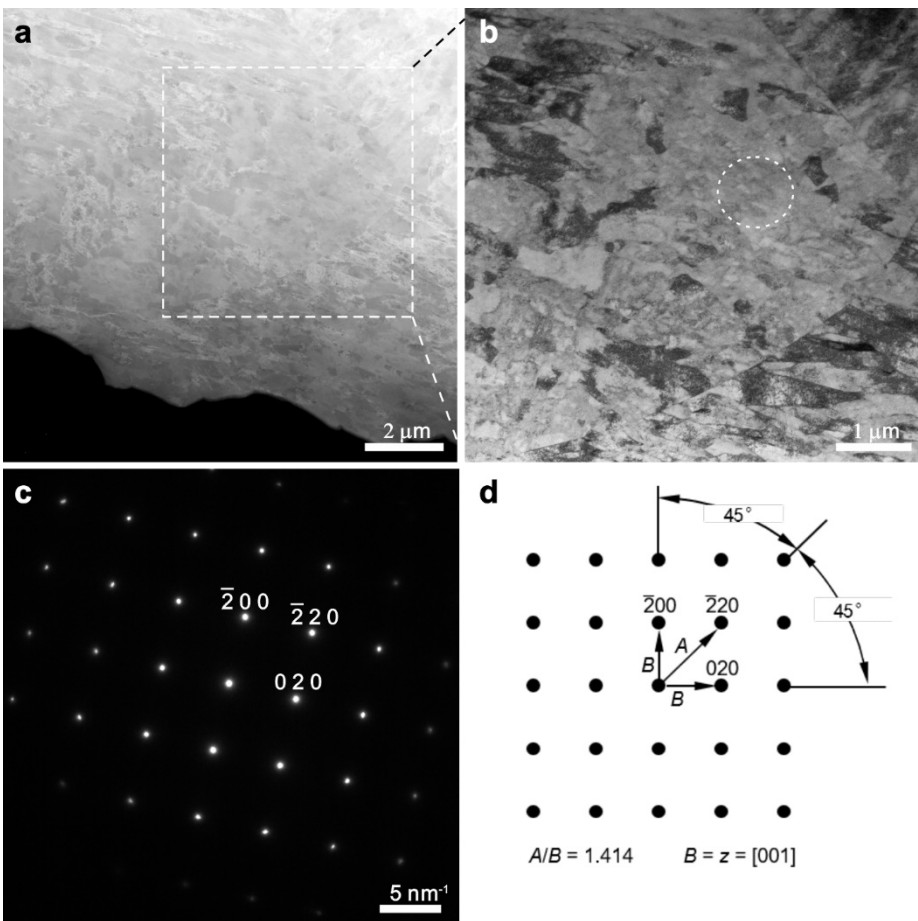

**Figure 3.** (**a**) Scanning transmission electron images and (**b**) bright field TEM images of the weld fusion line. (**c**) Selected area diffraction patterns of the area indicated by the white dashed circle in (**b**). (**d**) Standard diffraction patterns for face-centered cubic microstructure along the [0 0 1] axis. Note that the position of the TEM sample is indicated by the white dashed circle in Figure 1b. The white dashed square in (**a**) indicates the area for the bright field TEM image of (**b**). The white dashed circle indicates the area for selected area electron diffraction.

### 3.2. Mechanical Properties after Welding

Figure 4 shows the engineering stress–strain curves (ESSCs) of the base metal and the material after welding. The position of the tensile test samples for materials after welding is shown in Figure 1b. According to ESSCs, the yielding strength $\sigma_{0.2}$, tensile strength $\sigma_b$, and elongation to failure $\varepsilon_f$ of the base metal are 654 MPa, 1010 MPa, and 18.1%, respectively. In contrast, those of materials after welding are 571 MPa, 752 MPa, and 18.6%, respectively. After welding, the strength of the material decreased, but the ductility increased. The nominal tensile properties and microhardness of the welding repaired zone still exceed the required values ($\sigma_{0.2} \geq 550$ MPa, $\sigma_b \geq 750$ MPa, $\varepsilon_f \geq 15\%$, microhardness: 221–286) in related standard specifications [5,6]. 06Cr13Ni4Mo welding-repaired zones with FMPI induced by reticular austenite are acceptable and can be used as normal.

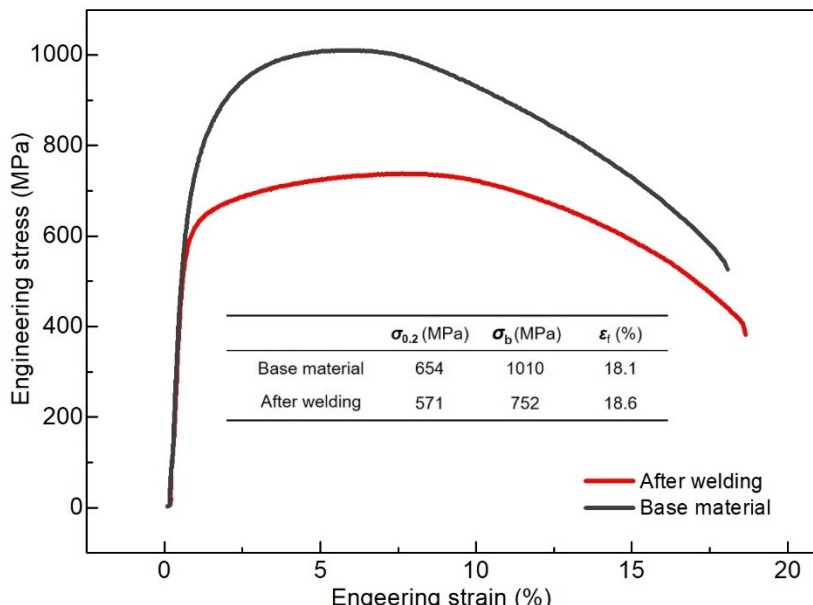

**Figure 4.** Tensile engineering stress–strain curves of the base metal and the materials after welding. Note that the position of the tensile test sample for materials after welding is indicated in Figure 1b.

As austenite is non-ferromagnetic [26], the fusion line reticular austenite reduces the apparent magnetic permeability, results in the leakage of magnetic flux, and further induces magnetic particle adsorption and FMPI [26]. In order to eliminate FMPI, it is necessary to explore the origin of the abnormal reticular austenite phase.

Due to the complexity of welding, several factors may be potential reasons for the abnormal reticular austenite formation, including the welding wire, the content of Ar and $CO_2$ of the shielding gas, the post-welding cooling rate, and so on. These factors were tested. However, it was found that these factors have no influence on the formation of abnormal reticular austenite. Considering that the main structure where reticular austenite presents is still tempered lath martensite(the same with the base metal), it was inferred that the abnormal reticular austenite may be induced by the thermal effect of welding on the base metal. Therefore, roasting tests of the base metal are performed to verify this inference.

*3.3. Microstructures of the Roasted Blocks*

Figure 5 shows the variation of microstructure and microhardness with the distance to the roasted top surface in the block. At a distance of 26 mm from the roasted surface, the microstructure is tempered lath martensite, the same as that of the base metal (Figure 5b). With the distance decreasing, the reticular white phase gradually appears on the martensitic cluster boundaries (Figure 5c), whose size, morphology, and distribution resemble that observed on the weld fusion line (Figure 1d,e). The mixed microstructure, consisting of reticular phase and tempered lath martensite, is mainly distributed over the range from 19 to 25 mm. For a distance of less than 19 mm, the reticular phase changes scarcely, but part of the martensite lath gradually transforms into coarse plates (Indicated by the arrows in Figure 5d).

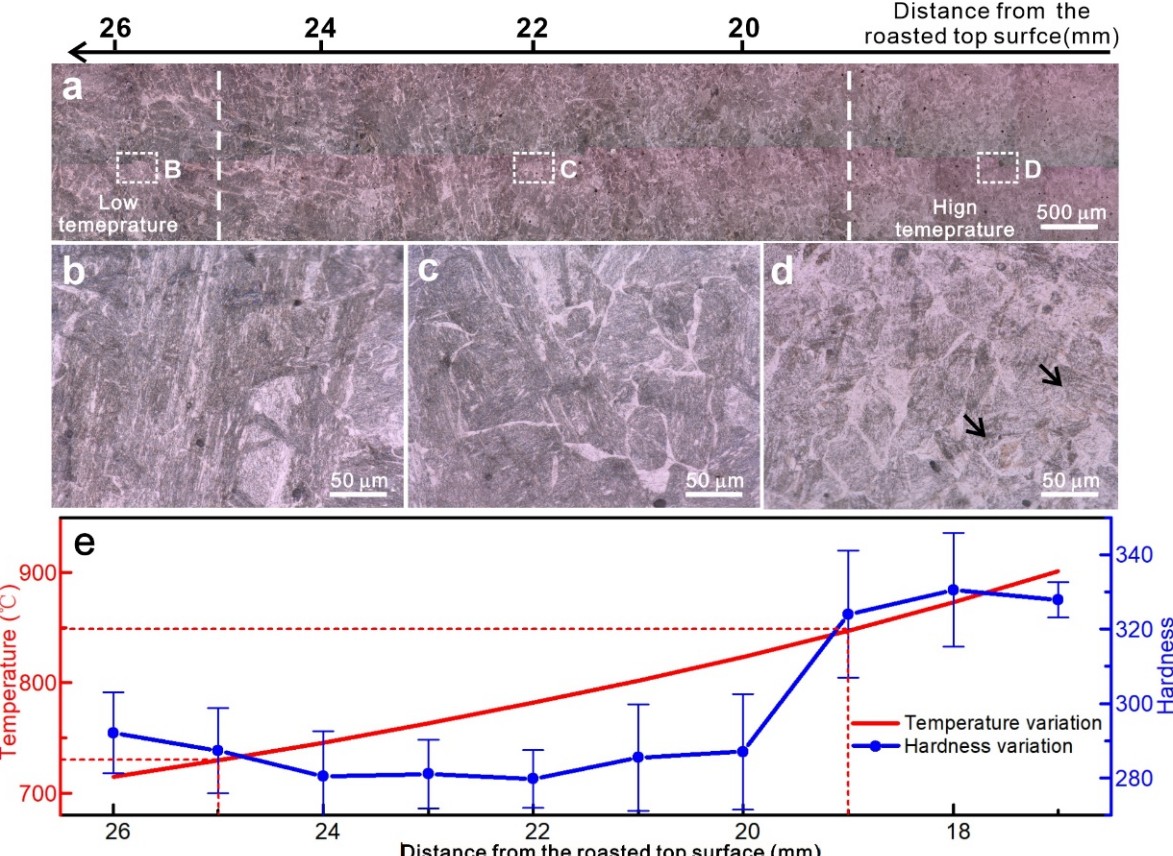

**Figure 5.** (**a**) Cross-sectional optical micrographs of roasted block. (**b–d**) Optical micrographs of the aeras indicated by (**B–D**) in (**a**). (**e**) Microhardness and the stable temperature variations with the distance from the roasted top surface.

With the distance to the roasted top surface decreasing, the microhardness of the roasted block first reduces and then increases. Over the range of 20–24 mm, the hardness is the lowest (279~285) and almost remains stable (Figure 5e). This distance range is in good consistency with that of the mixed microstructure (19–25 mm). Using Equation (2), the stable temperature during the roasting was calculated to be 847–729 °C over the range from 19 to 25 mm (Figure 5e).

Figure 6 shows the XRD patterns of the base metal and the roasted block. There is $\alpha'$ martensite and $\gamma$ austenite in both materials. However, compared with the base metal, the diffraction intensity of $\gamma$-austenite is greatly enhanced for the roasted material. This indicates the increase in the volume fraction of $\gamma$-austenite [33]. The relation between the diffraction peak intensity and phase volume fraction (Equations (3) and (4)) can be used to calculate the content of the $\alpha'$ martensite and $\gamma$ austenite [33]. By calculation, the volume fraction of austenite in the base metal was about 3.1 Vol.%, whereas it was 14.5 Vol.% after roasting. As the most significant change for microstructures induced by roasting is the formation of the reticular white phase, it can be deduced that the reticular white phase is $\gamma$ austenite.

$$V_\gamma = \frac{1/\mathrm{n}\sum_{j=1}^{n} I_\gamma^j / R_\gamma^j}{1/\mathrm{n}\sum_{j=1}^{n} I_\alpha^j / R_\alpha^j + 1/\mathrm{n}\sum_{j=1}^{n} I_\gamma^j / R_\gamma^j} \tag{3}$$

$$R = \frac{1}{v}F^2 P\left(\frac{1+\cos^2 2\theta}{\sin\theta\sin 2\theta}\right)e^{-2M} \tag{4}$$

where $n$, $I$, $R$, $\upsilon$, $F$, $P$, and $e^{-2M}$ are the number of diffraction peaks, the integral intensity, the material scattering factor, the single cell volume, the structure factor, the multiple factor, and the temperature factor, respectively [33].

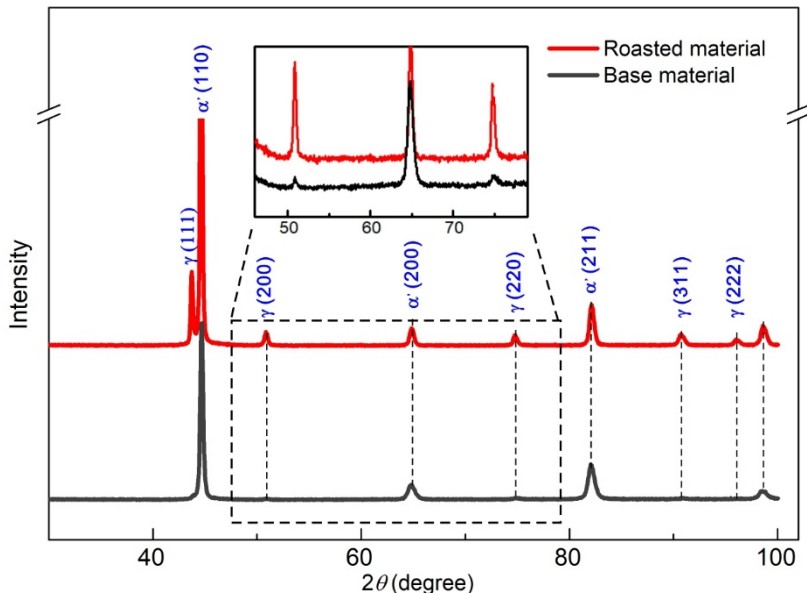

**Figure 6.** X-ray diffraction patterns of the base material and material from the roasted block. The position of the XRD sample for material from the roasted block is indicated by C in Figure 5a.

After roasting, reticular austenite presented in the base metal. Its morphology, size, and distribution resembled those of the reticular austenite on the weld fusion line. This verifies the inference that the reticular austenite on the fusion line is induced by the thermal effect of welding on the base metal.

## 4. Discussion

### 4.1. Influence of Reticular Austenite

As a major nondestructive examination method, post-welding FMPT is mainly used to detect material discontinuous defects, such as microcracks and pores [26,34]. As these discontinuous defects significantly deteriorate the mechanical properties of metals, nondestructive examination standard specifications require that welding-repaired zones with FMPI are unqualified [24,25]. However, FMPI in this study was caused by the non-ferromagnetic reticular austenite on the weld fusion line, which is completely different from that caused by the discontinuous welding defects.

Considering the position of the tensile test sample, its gauge region contained an inhomogeneous microstructure composed of weld fusion zone, weld fusion line, and heat-affected zone. During welding, the liquid-melted metal from the welding wire drops into the groove and cools down rapidly, which induces martensitic transformation. The microstructure of the fusion zone should be totally martensite. Since the strength of austenite is generally lower than that of martensite [4,27,28], the fusion line containing reticular austenite will have lower strength. This can be partially proved by the decrease in microhardness over the range of 19 to 25 mm in the roasted sample. Therefore, the decrease in the yield strength and tensile strength of materials after welding should be mainly caused by the reticular austenite. Even then, however, its nominal mechanical properties, including the tensile properties and the microhardness, still exceed the required values in related standard specifications [5,6]. 06Cr13Ni4Mo welding-repaired zones with FMPI induced by reticular austenite are acceptable. On the other hand, though, it should be noted that as Ni can stabilize the austenite, the volume fraction of reticular austenite on the weld fusion line will further increase if more Ni is added to the steel. More reticular austenite will further decrease the strength of the welding-repaired zone. For other low-carbon Cr-Ni

martensitic stainless steels, such as 06Cr13Ni5Mo and 04Cr13Ni5Mo, the tensile strength may drop to below 750 MPa, which will make them unqualified. This special case of other low carbon Cr-Ni martensitic stainless steels should be fully taken into consideration when FMPTs are performed for the examination of welding-repaired zones in produced castings.

*4.2. Formation Mechanism of Reticular Austenite*

Similar to the reversed austenite transformation during low-temperature tempering [4,9,11,35], the reticular austenite found in this study is caused by the thermal effect of welding but at a higher temperature (729–847 °C). Previous studies have proved that reversed austenite forms through the diffusion-controlled mechanism [9]. Austenite stabilizing Ni atoms diffuses at defects (dislocations, boundaries, interfaces), resulting in Ni enrichment and promoting austenitizing at the low temperature of tempering [9,36]. Due to stabilization by Ni, martensite transformation cannot occur for the reversed austenite during cooling. Thus, reversed austenite is retained. The higher the Ni content, the lower the $M_s$ point. In addition to boundaries between martensitic laths, other typical defects in 06Cr13Ni4Mo include the martensitic cluster boundaries. As the width of martensitic lath obtained by quenching is only 100 nm, there are plenty of lath boundaries, and Ni atoms diffuse to these boundaries by short-range diffusion. As the density of martensitic cluster boundary is much lower than that of martensite lath boundary, Ni atoms have to diffuse for a long range to reach those boundaries, and the content will be much lower than that at the lath boundary (Figure 2b). Thus, the reticular austenite formation temperature at martensitic cluster boundaries (729–847 °C) is much higher than that at lath boundaries (400–600 °C). This may also be the reason why reticular retained austenite was not found in low-temperature tempered 06Cr13Ni4Mo.

In practical production, to avoid post-welding cracking, the welding-repaired zone usually cools down at a much lower speed (due to air cooling instead of forced wind cooling). The low cooling speed, together with the stabilizing effect by Ni, makes the formed reticular austenite on the weld fusion line stable after cooling down. This is why reticular retained austenite formed on the weld fusion line and induced FMPI in 06Cr13Ni4Mo welding-repaired zones.

## 5. Conclusions

Inspired by FMPI, a novel reticular phase distributed along the boundaries of martensitic clusters was found on the weld fusion line of 06Cr13Ni4Mo. In terms of its influence on FMPT result and mechanical properties of the welding repaired zone, the following conclusions can be drawn:

(1) The reticular phase on the weld fusion line was induced by the thermal effect of welding on the base metal. It is one kind of retained austenite.

(2) Due to its non-ferromagnetic nature, reticular austenite induces FMPI on the weld fusion line of 06Cr13Ni4Mo.

(3) Though reticular retained austenite decreases the mechanical properties of the welding-repaired zone, the yield strength $\sigma_{0.2}$, tensile strength $\sigma_b$, elongation to failure $\varepsilon_f$, and microhardness of the welding joint are 571 MPa, 752 MPa, 18.6%, and 279 MPa, respectively, and still exceed the required values in related standard specifications.

(4) 06Cr13Ni4Mo welding-repaired zones showing FMPI induced by reticular retained austenite should not be judged to be unqualified. This special case of reticular retained austenite-induced FMPI should be taken into consideration when performing FMPT on 06Cr13Ni4Mo welding-repaired zones.

**Author Contributions:** F.P. performed the welding test, heat treatments, and tensile tests; Y.Z. carried out hardness tests and TEM analysis; Z.F. performed XRD. J.L. provided scientific guidance and supervised the whole investigation process. All the authors discussed the results and contributed in writing the paper. All authors have read and agreed to the published version of the manuscript.

**Funding:** The research was funded by the the Natural Science Foundation of Anhui Province (2108085QE189), Talent project of Anhui Polytechnic University (S022020004, KZ42020085), Industry and university jointed research project of Fanchang County (2021fccyxtb2, 2021fccyxtb3), Key research and development program of Ningxia Province (2022ZDYF0619).

**Institutional Review Board Statement:** Not applicable.

**Informed Consent Statement:** Not applicable.

**Data Availability Statement:** All data that support the findings of this study are available from the corresponding author upon reasonable request.

**Acknowledgments:** The authors acknowledge the financial support by the Natural Science Foundation of Anhui Province (2108085QE189), Talent project of Anhui Polytechnic University (S022020004, KZ42020085), Industry and university jointed research project of Fanchang County (2021fccyxtb2, 2021fccyxtb3), Key research and development program of Ningxia Province (2022ZDYF0619).

**Conflicts of Interest:** The authors declare no conflict of interest.

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
