# Peer review of "A Novel Reticular Retained Austenite on the Weld Fusion Line of Low Carbon Martensitic Stainless Steel 06Cr13Ni4Mo and the Influence on the Mechanical Properties"

_metals, doi:10.3390/met12030432_

Round 1
Reviewer 1 Report
This study investigated the phase identification and mechanical properties due to reticular austenite formation in fusion zone. Minor revisions are required as follows:
- line 29: Microhardness => microhardness
- line 45: different => various
- line 80 and other equations: Equ. 1 => Eq. 1
- line 134: mechanical => Mechanical
- Figure 3. The gauge region of dog-born tensile specimens contained an inhomogeneous microstructure including fusion zone and heat affected zone. It is required to discuss the inhomogeneous microstructure effect on the tensile behavior.
Reviewer 2 Report
In this manuscript, the authors analyzed the microstructures of the 06Cr13Ni4Mo welding fusion line. They considered the formation mechanism of a novel reticular phase that affects the mechanical properties of 06Cr13Ni4Mo welded joints.
The article contains an interesting practical aspect concerning the possibility of using the analyses carried out to examine 06Cr13Ni4Mo welding joints.
The article's subject contains interesting observations and may arouse interest, which justifies the publication of the article. The work has the correct structure, but a small number of tests carried out raises doubts about the repeatability of the results. The presented results are well-illustrated, and only some descriptions contain shortcomings that the authors should correct. Therefore, it is recommended to make minor adjustments before publication.
Please read the detailed notes in the attached file.

Reviewer 3 Report
English and welding terminology needs to be corrected (e.g. …. they still meet the …, labelled, gradient heat treated).
In the introduction, you need to write about what diagnostic methods are used for stainless steels.
Line 62 - what heat treatment state.
Blocks should be replaced with sheets.
It is necessary to present the workpieces before welding, the gap, cutting edges, etc. on the figure.
It is need to specify how many welding passes were there?
What is shown in figure 1a?
Line 92 - the loading speed is incorrect.
For the convenience of the reader, it is better to present a macrosection of the entire welded joint.
In my opinion, what the authors call "reticular austenite" is usually called “retained austenite”.
Round 2
Reviewer 3 Report
-
Author Response
Thank you for reviewing.